# Strong optical anisotropy in one-dimensional phosphorus wavy tubes

Shuai Zhang [1,2,15], Zhaolong Liu[3,4,15], Tongtong Jiang[1], Chen Wang[1,5], Jiahui Wang[6], Han Wang[5], Minqiang Fan[1], Li Yang[1,4], Yang Li[1], Liping Ding[7,8] ✉, Ying Yu[9], Xiaodong Hao[6], Shufang Ma[6], Bingshe Xu[6], Xiaolong Chen [5], Cong Ye[2], Xianfeng Chen [10], Paul K. Chu [11], Shifeng Jin [3,4], Feng Ding [8] ✉, Xue-Feng Yu [1,4,12] ✉, Zhipei Sun [13] ✉ & Jiahong Wang [1,4,12,14] ✉

Anisotropic materials with intrinsic one-dimensional architectures, where chains or tubes align along a crystallographic axis, exhibit direction-dependent optical responses and serve as ideal building blocks for polarization-sensitive optoelectronics. While progress exists in engineered compounds, discovering elemental crystals with naturally ordered one-dimensional building blocks exhibiting giant optical anisotropy remains challenging. Here, we report the synthesis of a direct-bandgap semiconducting one-dimensional phosphorus single crystal composed of unique wavy polygonal tubes. The monoclinic lattice structure is revealed by single-crystal X-ray diffraction and advanced transmission electron microscopy. The crystal exhibits giant birefringence in the visible and near-infrared regions, stemming from electron localization and anisotropic transitions of the phosphorus $3p$ orbital along the tube axis. The low-symmetry structure endows remarkable linear and nonlinear optical anisotropies, including orientation-dependent photoluminescence, Raman scattering, and second-harmonic generation. This study establishes a paradigm for designing giant optical anisotropies, opening avenues for on-chip polarization devices and nonlinear photonic circuits.

Optical anisotropy, governing polarization-dependent light-matter interactions, serves as the cornerstone for advanced integrated photonic technologies such as on-chip polarizers, electro-optic modulators, and directional optical communication[1-3]. Therefore, there is a growing demand for materials with strong anisotropic optical responses. Traditional anisotropic crystals (such as calcite and rutile $TiO_2$) are limited by the symmetry constraints of the atomic structure, with an in-plane birefringence generally below 0.3[4,5], making it difficult to meet the requirements of miniaturized integrated devices. Low-dimensional materials boasting adjustable electronic confinement effects and structural degrees of freedom provide a new approach to overcoming the hurdle[6-9]. Although low-dimensional anisotropic

materials from layered GeS (birefringence $\Delta n \approx 0.5$) and black phosphorus ($\Delta n \approx 0.3$) to $ReS_2$ have been explored[10-14], it is difficult to satisfy the demand for high polarization selectivity since the planar structure restricts its optical anisotropy.

Latest theoretical studies propose that one-dimensional (1D) architectures can amplify optical anisotropy through symmetry-breaking building blocks, whose chains or tubes highly oriented along the main axis can amplify the optical response differences through the synergy of chemical bond anisotropy and electron-phonon coupling effects[15,16]. Preparing high-quality 1D single-crystal materials with intrinsically strong anisotropy through atomic-level structural design has become one of the main challenges in the current

A full list of affiliations appears at the end of the paper. ✉e-mail: scu_ding@163.com; f.ding@siat.ac.cn; xf.yu@siat.ac.cn; zhipei.sun@aalto.fi; jh.wang1@siat.ac.cn

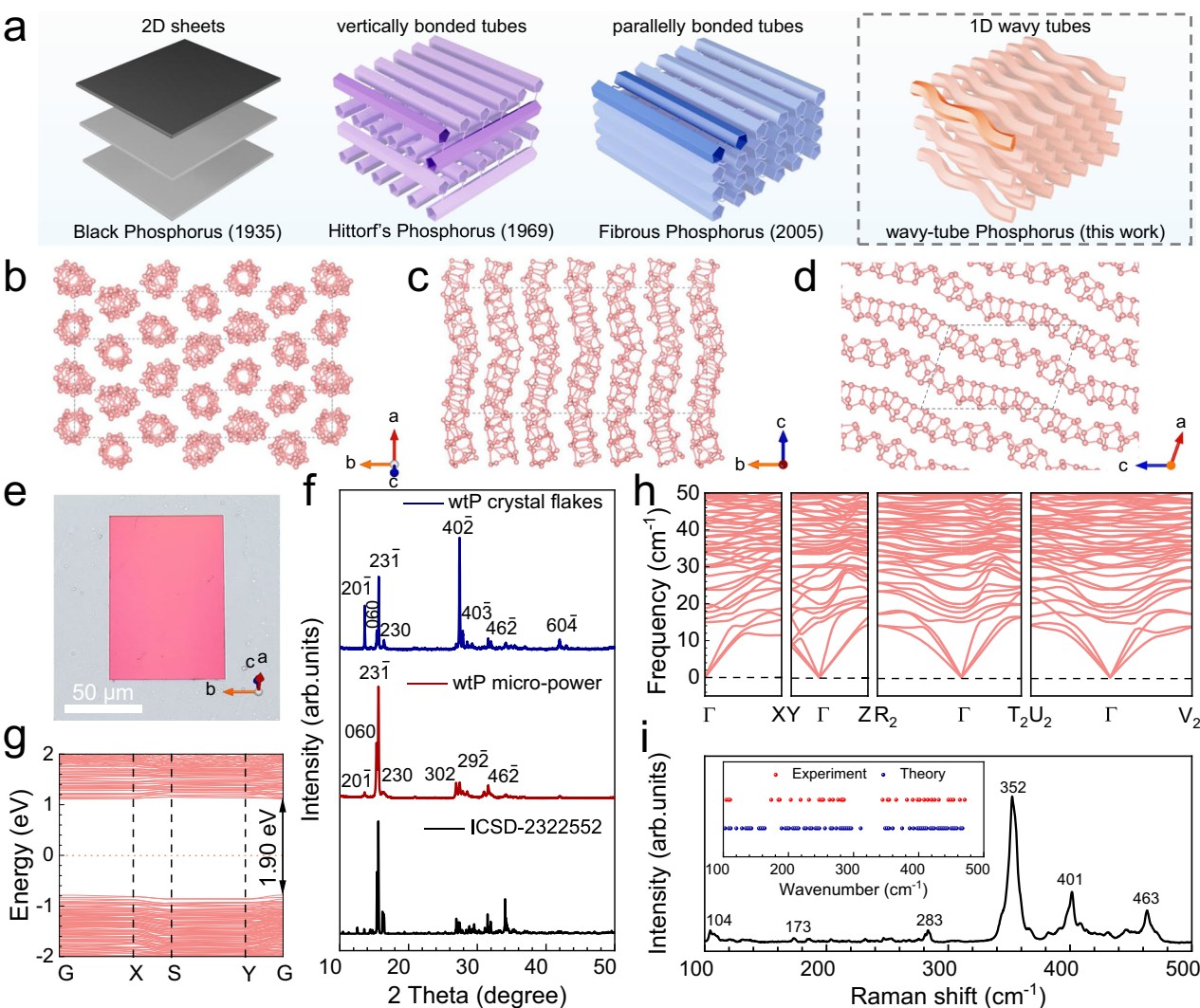

**Fig. 1 | Crystal structure of wavy-tube phosphorus single crystals. a** Structures of typical phosphorus allotropes: 2D black phosphorus composed of puckered sheets[27], 2D Hittorf's phosphorus composed of vertically covalently straight tubes[28], quasi-1D fibrous phosphorus composed of parallelly covalently straight bi-tubes[29], and 1D wavy-tube phosphorus (wtP) composed of wave-shaped polygonal tubes. **b** View of the wtP crystal structure along the [102] direction, confirming the tube character. **c** View of the wtP crystal structure along the *a*-axis, revealing six parallel tubes within a single unit cell with a slightly different configuration. **d** View of the wtP monolayer crystal structure along the *b*-axis, indicating the repeating unit of an individual tube is [P10]P2[. **e** Optical microscopic photograph of wtP single crystal in non-polarized light. **f** X-ray diffraction (XRD) patterns of fresh wtP single crystal flakes with the preferred orientation (blue line), ground wtP micro-powder without the preferred orientation (red line) and stimulated XRD pattern of wtP (black line). **g** Electronic band structure of the calculated wtP unit cell, where G, X, S, and Y denote the critical k points in the Brillouin zone. **h** Phonon scattering curves for wtP. **i** Unpolarized Raman spectrum of an individual wtP flake. The inset is a comparison between the experimental energy (red) and the calculated phonon energy (blue).

optoelectronic material field. Notably, a birefringence of Δn ≈ 0.76 in the mid- and long-wave infrared region has been achieved from BaTiS₃ nanowires through the directional arrangement of [TiS₆] octahedral chains, revealing the potential of chains for anisotropic engineering[17]. In addition to compounds, experimental validation of elemental materials has been realized in elemental tellurium plates and fibrous phosphorus allotropes[18–21]. However, the optical response difference is weak due to the covalent coupling between adjacent straight tubes of fibrous phosphorus[22,23]. While the $3d$ or more complex outer layer orbit hybridization primarily enables large optical anisotropy (Δn > 0.5), there are limited cases where $3p$ orbit hybridization plays a dominant role in low-dimensional nonlinear crystals. Generally, overcoming inter-tube cross-linking and improving intra-tube atomic arrangement order are challenges for disrupting the rotational symmetry and enhancing optical anisotropy.

Long predicted but never seen, Type-II red phosphorus is thought to consist of wavy, tubular chains whose broken inter-tube symmetry (Fig. 1a and Supplementary Fig. 1) should endow the crystal with extraordinary optical anisotropy[24–26]. However, since its initial discovery by Roth et al. in 1947, the determination of its precise crystal structure has remained a persistent and formidable challenge in the field. The core obstacle has been the extreme difficulty in obtaining high-quality single crystals suitable for atomic-resolution structural analysis. Here, we isolate the first single-crystalline specimens - 100 micrometer-scale crystals grown by a tailored chemical vapor transport (CVT) route and resolve their lattice of periodically stacked wavy polygonal tubes. This definitive structural proof unlocks the material's long-anticipated properties: the unique bonding framework yields record polarizability contrast and striking linear and nonlinear anisotropies in second-harmonic generation and photoluminescence. Our discovery not only settles a decades-old structural question but also establishes Type-II red phosphorus as a new platform for directionally engineered photonics.

## Results

### Synthesis of 1D phosphorus crystals composed of wavy tubes

Single crystals of the 1D phosphorus allotrope are synthesized by modified CVT in a two-zone tube furnace with amorphous red phosphorus (aRP) and tin as precursors. The micro-thin orange-red flakes are obtained in the cold zone after one week. Historical problems in the preparation of high-quality type II red phosphorus single crystals have been fundamentally solved through a synergistic innovation strategy of precise thermal control, mineralizer revolution, kinetic regulation and precursor engineering (discussions in Supplementary Section 1). Detailed synthesis and temperature profiles are available in the experimental section of the methods and Supplementary Fig. 2. Rectangular plates, predominantly appearing orange-red in the transmission model and orange-yellow in the reflection model (Fig. 1e and Supplementary Fig. 3), are consistent with the previously reported color of Type-II red phosphorus. The atomic structure of the orange-yellow crystal is determined through single-crystal X-ray diffraction (SCXRD) to be monoclinic, with a lattice structure in space group $P2_1$ (Supplementary Table 2) and lattice constants of $a = 13.0518(3)$ Å, $b = 34.4922(4)$ Å, $c = 18.8538(4)$ Å, and $\beta = 109.737(3)°$. The unit cell atomic views in Fig. 1b–d reveal twelve 1D polygonal tubes aligned parallelly along the [102] direction, with each tube consisting of 30 atoms forming a V-shape wavy structure, therefore, this material is named as wavy-tube phosphorus (wtP). An individual tube has a characteristic wave periodicity of 17.7 Å with a bending angle of about 150°. Despite variations of the starting configurations contributing to the unique visual structure of the polygonal tubes, the smallest repeating unit of an individual tube is [P10]P2[(as shown in Supplementary Fig. 1). In contrast to the vertically or parallelly covalently bonded straight tubes observed in Hittorf's phosphorus (HP) and fibrous phosphorus (FP) (Fig. 1a and Supplementary Fig. 1)[23,28], the adjacent wavy tubes of wtP are mutually independent. X-ray diffraction (XRD) is conducted on the single-crystal regular flake samples (the blue line in Fig. 1f). The sharp diffraction peaks corresponding to the {20-1}, {060}, {23-1}, and {40-2} are located at 13.6°, 15.4°, 15.6°, and 27.3°, respectively, indicating the preferred exposed oriented of the wtP flakes. Furthermore, in the XRD pattern (the red line) of the ground wtP micro-powders (PXRD) without preferred orientation, the {23-1} and {060} planes exhibit the highest intensity in the non-oriented PXRD pattern due to their extinction coefficients in these tightly packed orientations. The PXRD results match the stimulated standard plots derived from the lattice structure (ICSD No. 2322552, the corresponding cif file can be found in Supplementary Information).

A direct bandgap of about 1.90 eV at the G point in the Brillouin zone is predicted based on the density-functional theory[30], and the p-type semiconductor property is demonstrated at the same time (Fig. 1g). The calculated phonon dispersion curves in Fig. 1h show good agreement between phonon energies and experimental results (Fig. 1i)[31]. Diverse Raman scattering modes are observed from the 1D wavy polygonal wtP due to the low symmetry of the crystal structure and the massive atoms in a unit cell. Both the XRD pattern and Raman scattering peaks of wtP closely match those of Type-II red phosphorus reported by Roth et al.[32] (discussions in Supplementary Section 3). Despite extensive efforts to identify the tubular packing tube with wavy patterns of Type-II phosphorus, the complete lattice structure remains elusive due to the nano-scale samples[24–26,32–34]. By utilizing a bulk single-crystal flake instead of nanowires, the crystal structure of wtP (Type-II phosphorus) is directly identified for the first time in this study.

### Atomic structure of 1D phosphorus wavy tubes

To further determine the crystal structure of wtP and establish a solid foundation for the optical analysis, atomically resolved high-angle annular dark field-scanning transmission electron microscopy (HAADF-STEM) is employed. As shown in Fig. 2a, the TEM samples with

different crystal orientations are prepared by a focused ion beam (FIB), and the areas are labeled I, II, and III, respectively. The cross-sectional surfaces of the samples obtained by FIB cutting exhibit uniformity and integrity (Fig. 2b), while the corresponding elemental maps confirm a single element of phosphorus (Supplementary Fig. 7). Before capturing the TEM images, minor position adjustments are made to align the zone axis parallel to the electron beam. The HAADF-STEM image of cross-section area I (Fig. 2d) reveals that a tight packing arrangement of polygonal tubes forms a quasi-hexagonal network, where the variation of edge lengths is attributed to the tube waviness. According to the selected-area electron diffraction (SAED) pattern (Fig. 2e), the lattice distances are measured as 0.56 nm, 0.56 nm, and 0.57 nm, whose lattice plane angles are $\angle_{12} = 57.7°$, $\angle_{23} = 61.9°$, which correspond to the (-2-31), (-231), and (060) planes of the wtP, respectively. The results indicate that the polygonal phosphorus tubes of the wtP extend along the [102] direction. Therefore, the atomic structure model of the simulated (102) plane shows good agreement with the experimental image. Despite the tubes being wavy rather than completely straight, the STEM image highlights the independent and non-interconnected characteristics of each tube.

When the electron beam is transmitted through the wtP crystal from another in-plane orientation, alternating bright and dark parallel stripes with a lattice spacing of 0.65 nm are observed by STEM (Fig. 2f), which are indexed as the lattice plane of (-201) and the crystal orientation of [010] in the SAED pattern (Fig. 2g). Besides, sub-diffraction points appear from the SAED pattern due to the superlattice-like arrangement. Furthermore, in the STEM image of area III (Fig. 2h), the wavy tube structure exhibits a repeated V-shape structure with a unit length of 1.77 nm corresponding to the (001) plane. Meanwhile, the 1.14 nm spacing corresponding to the (030) plane is determined vertically. Based on the aforementioned analysis, the 3 $d$ model of the wtP single crystal plate is simulated in Fig. 2c, in which the three primary zone axes are [102], [010], and [100], while the corresponding spatial angles are $\angle_{[102][010]} = 90°$, $\angle_{[010][100]} = 90°$, and $\angle_{[102][100]} = 63°$. Since the predominant growth direction [102] (defined as $c^*$-axis) is perpendicular to [-201] and [010], the XRD pattern of the wtP single plates displays distinct preferred diffraction peaks corresponding to the {-201}, {060}, and {-402} planes.

### Strong in-plane birefringence of 1D phosphorus wavy tubes

Elucidation of the wtP crystal structure reveals a unique architecture, with a one-dimensional covalent chain confined within a van der Waals stacking arrangement. Intrinsic anisotropy inevitably influences macroscopic optical properties. As shown in Fig. 3, optical constants of wtP are extracted employing ellipsometry (details are provided in the experimental section of the methods and Supplementary Section 5). The refractive index $n$ exhibits significant variation along the in-plane $b$ and $c^*$ axes within the 450–800 nm wavelength range (Fig. 3a), indicating a substantial in-plane birefringence (Fig. 3b). This finding also is supported by polarization-resolved optical microscopy under cross-polarized illumination (Supplementary Section 7). The wavelength-dependent refractive index, extinction coefficient, birefringence, and linear dichroism parameters are well agreement with the complete optical response characteristics elucidated by density functional theory calculations (Fig. 3c, d, Supplementary Fig. 9). Combined with its biaxial crystal properties determined by the monoclinic characteristics, calculations indicate that wtP crystals exhibit notable birefringence across the $ac^*$ and $bc^*$ planes (discussions in Supplementary Section 6). This suggests that optical device designs based on wtP offer high flexibility, eliminating the critical crystal orientation selection. Experimental observations confirm wtP exhibits giant birefringence ($\Delta n \approx 0.95@450$ nm, $0.75@600$ nm, and $0.58@800$ nm) in the visible and near infrared region, outperforming typical anisotropic materials (Fig. 3e) and even surpassing the newly reported engineered compound $C_3H_8N_6I_6 \cdot 3H_2O$ within the visible band, highlighting the

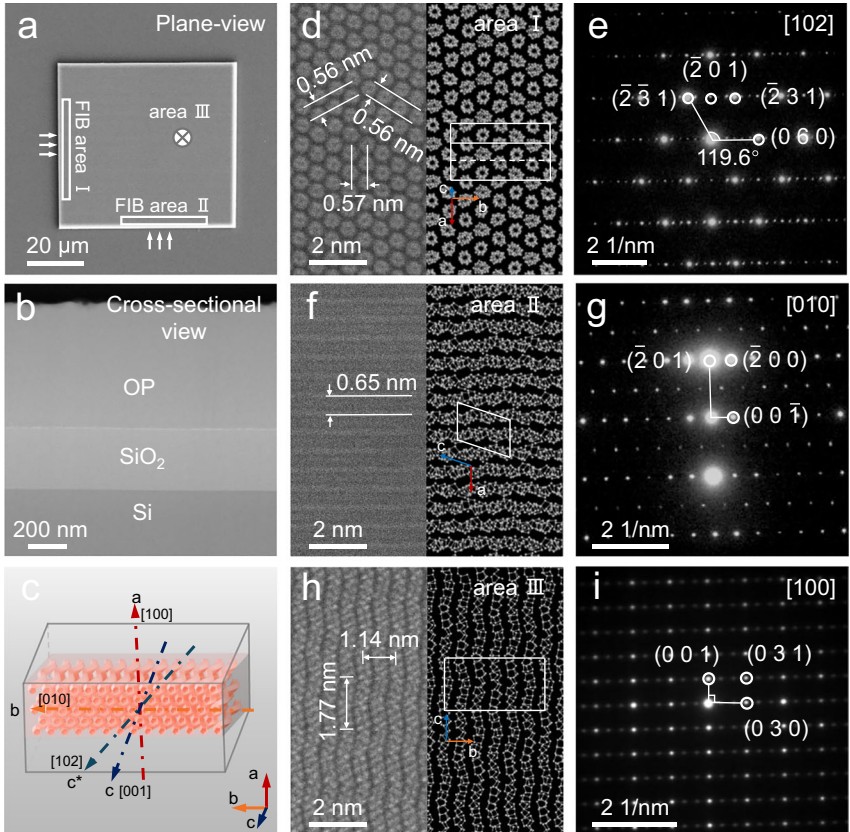

**Fig. 2 | Structural characterization of wavy-tube phosphorus crystals.**
**a** Scanning electron microscopy image of the wavy-tube phosphorus (wtP) single crystal flake used for transmission electron microscopy (TEM) characterization with two in-plane direction thin cross-section samples fabricated by focused ion beam (FIB) labeled as area I and FIB area II and out-plane direction sample defined as area III. **b** Low-magnification high-angle annular dark field-scanning transmission electron microscopy (HAADF-STEM) image of the cross-sectional area I. **c** Modeling diagram of a typical wtP flake composed of wavy polygonal tubes and four labeled crystal orientations. **d**, **e** HAADF-STEM image and the simulated atomic graph, as well as **e** selected-area electron diffraction (SAED) pattern of the cross-section area I calibrated with the view direction of [102]. **f**, **g** HAADF-STEM image and the simulated atomic graph and **g** SAED pattern of the cross-section area II calibrated with the view direction of [010]. **h**, **i** HAADF-STEM image and the simulated atomic graph and **i** SAED pattern of the area III calibrated with the view direction of [100].

decisive role of the one-dimensional tubular structure in conferring optical anisotropy[4,5,7,8,21,35–40]. Meanwhile, the absolute refractive index $n$ of wtP is larger than 3.5 in the band edge and lager than 3.0 in the whole infrared region (E < 1.0 eV). With a calculated bandgap ($E_g$) of 1.9 eV, wtP is identified as a super-Mossian material, meeting the criterion $n^4 \cdot E_g \gg K$ ($K \approx 100$)[41]. Compared to the predicted $n$ value of 2.69 based on Moss rule, this represents an increase of approximately 30% in the visible range, indicating a marked enhancement in the material's ability to confine light[42]. This enhancement is attributed to the flat valence band maximum and conduction band minimum of wtP, leading to a high density of states and reinforcing light-induced polarization[43]. A high refractive index enables the creation of micro-resonators with strong optical confinement effects, substantially enhancing light emission efficiency. It is also crucial for fabricating nanoscale waveguides, modulators, and optical switches, enabling smaller optical component sizes and higher integration density on chips.

In an attempt to elucidate the intrinsic connection between macro-optical properties and micro-electronic structure, a theoretical analysis is carried out based on the electron localization function (ELF) and the partial density of states (PDOS)[44,45]. As shown in Fig. 3f, the three-dimensional visualization of the ELF reveals an isosurface of 0.9 along the [102] direction, with regularly arranged charge distributions surrounding the phosphorus tubes. The charge localization distribution closely aligns with the 1D tubular morphology, which is observed through the 2D ELF projection of the (100) crystal plane (Fig. 3g). Benefiting from the isolated tube structure without covalent cross-

linking, the charge exhibits tightly localized states, forming an aniso-tropic charge distribution that induces giant birefringence in the $bc^*$ plane for wtP. Moreover, the electronic band structure near the Fermi level, combined with PDOS analysis, reveals a dominant contribution from $3p$ orbitals with negligible $s$ orbital effects, reflecting strong orbital anisotropy (Fig. 3h). In other words, the optical anisotropy of the wtP stems from the directional alignment of the P $3p$ orbitals within its wrinkled pentagonal tube, which contrasts with engineered compound materials that rely on 1D iodine chain alignments or 2D confined excitons that require complex band engineering[7,8,17,40]. It highlights the unique potential of elemental crystals to achieve high intrinsic anisotropy, and reveals a physical model guided by 1D structural motifs to realize large refractive index and universal giant anisotropy within bulk crystals.

## Optical anisotropy of 1D phosphorus wavy tubes

The 1D wavy polygonal tube feature of wtP defines its linear optical properties as well as induces an abundance of anisotropic optical phenomena. Raman scattering reveals a direct connection between the lattice symmetry and the structure characteristics of tubes. Figure 4a shows the polarized Raman spectrum of a wtP single-crystal flake. The polarization angle is defined as the angle between the incident light polarization direction and the c*-axis (also known as the [102] direction in reciprocal space). As shown in Fig. 4b, the polar plots of polarized Raman intensity for the wtP crystal in co- and cross-polarized configurations display maximum intensities at 0°/180° and 45°/225°, respectively. The considerable anisotropy in the Raman peak is

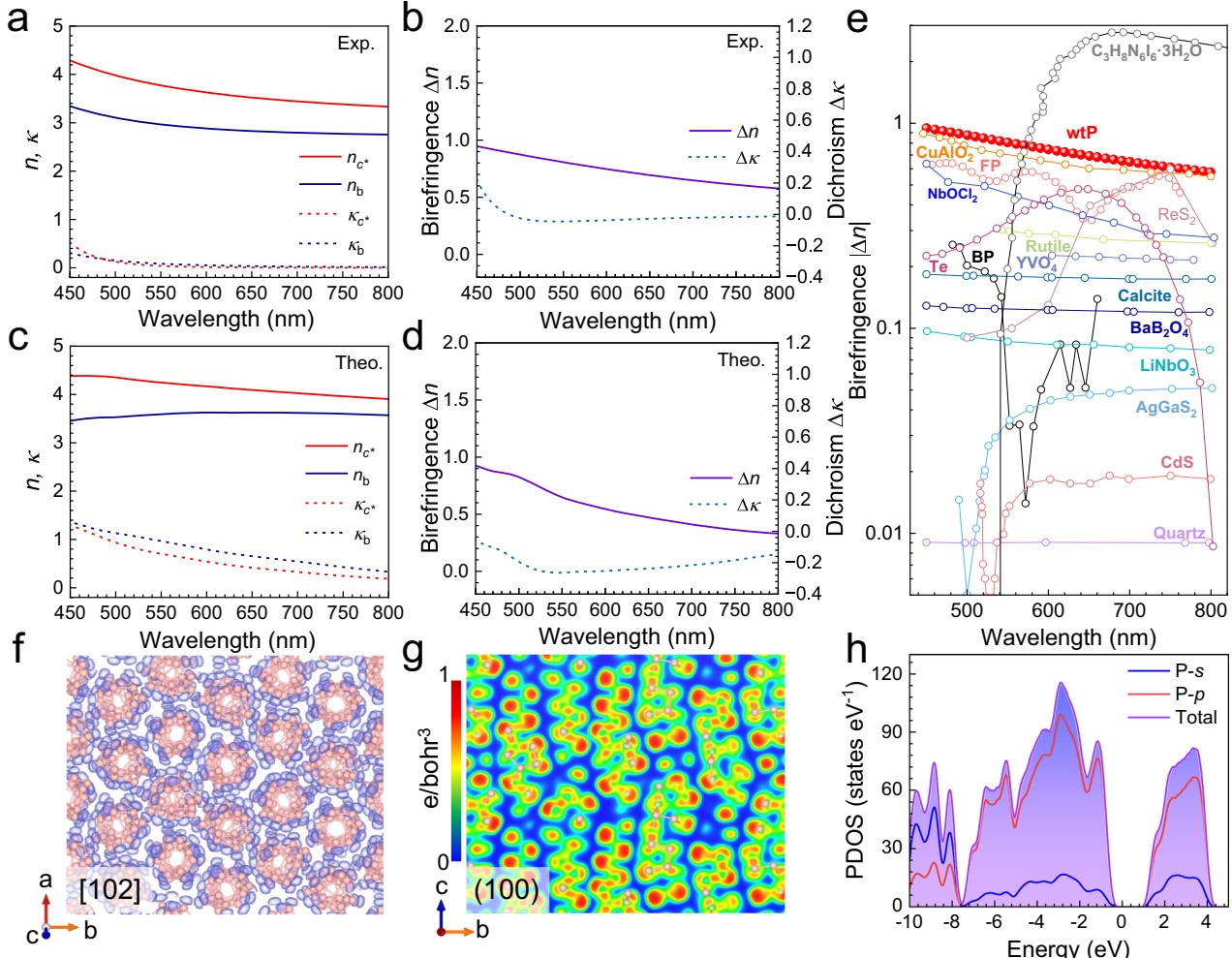

**Fig. 3 | Strong in-plane birefringence of wavy-tube phosphorus.**
**a** Experimentally measured refractive index ($n$) and extinction coefficient ($\kappa$) values along the in-plane $c^*$ and $b$-axis directions. **b** Birefringence ($\Delta n$) and dichroism ($\Delta \kappa$) derived from the experimental data in **a**. **c** Theoretically calculated $n$ and $\kappa$ values along the in-plane $c^*$ and $b$-axis directions. **d** Birefringence ($\Delta n$) and dichroism ($\Delta \kappa$) derived from the data in **c**. Exp. (Theo.) denotes experimental (theoretical) data. **e** Comparative analysis of absolute birefringence values of previously reported birefringent materials and wavy-tube phosphorus (wtP)[4,5,7,8,21,35–40]. **f** Three-dimensional visualization of the electron localization function (ELF) with an iso-surface value of 0.9 for wtP. The blue shade indicates the charge density distribution around the phosphorus atoms. **g** Projection of the ELF on (100) crystal plane for wtP. The iso-values increase from blue to red, and the maximum intensity is normalized to 1. **h** Calculated PDOS of wtP, including the P 2 $s$ orbital, P 2 $p$ orbital and the total contribution.

observed for all vibrational modes (Supplementary Fig. 12), with intensity distribution strongly correlated with the crystal orientation. Owing to the disrupted triple rotational symmetry, anisotropic non-linear optical responses are generated by the wtP flake[20,46].

Under 800 nm laser excitation with varying power densities, second-harmonic generation (SHG) signals at 402 nm are detected (Supplementary Fig. 13). The logarithmical plot of emission intensity against excitation power reveals a slope of 2.02, consistent with the theoretical value of 2. Furthermore, polarization-resolved SHG confirms the strong in-plane anisotropy (Fig. 4c and Supplementary Table 4). The polar graph in Fig. 4d shows a two-lobed distribution in both the parallel-polarized and vertical configurations, which are closely associated with the 1D tube in wtP. Besides, the anisotropic semiconductor property of the wtP flake is studied by photoluminescence (PL), as shown in Fig. 4e, the 650 nm light emission is excited by a 532 nm laser. The p-type band structure of the wtP crystal is further determined by ultraviolet photoelectron spectroscopy (Supplementary Fig. 14). Polarization-dependent phenomena are found in the PL spectra, as shown in Fig. 4f. The maximum intensities obtained in the co-polarized or cross-polarized configurations appear at 0°/180° or 90°/270°, respectively. The linear dichroism defined by

$\rho = \frac{I_{co} - I_{cross}}{I_{co} + I_{cross}}$ is used to quantify the magnitude of PL anisotropy, with a value of 86% surpassing that of other 2D Van der Waals materials (Supplementary Fig. 15). The bright and orientation-dependent PL characteristics of wtP suggest its promising potential for polarized photovoltaic and optoelectronic applications[16,47,48].

## Discussion

1D wtP single crystals are synthesized by a modified CVT technique to resolve the long-debated atomic structure of Type II red phosphorus as wavy polygonal tubes with a unique $P2_1$ monoclinic lattice structure. The 1D tube structure amplifies the directional polarizability differences and produces giant birefringence in the visible and near infrared region. The strong 3$p$-orbital hybridization is determined as the origin of the flat band structure and the polarization-sensitive linear and nonlinear optical properties. Upon excitation by polarized light parallel and perpendicular to the $c^*$-axis of wtP, substantial variations in the Raman, PL, and SHG intensities are observed. These breakthroughs establish wtP as benchmark materials for polarization optics and present a design blueprint based on tubes for the engineering of extreme optical anisotropy and optical confinement effects.

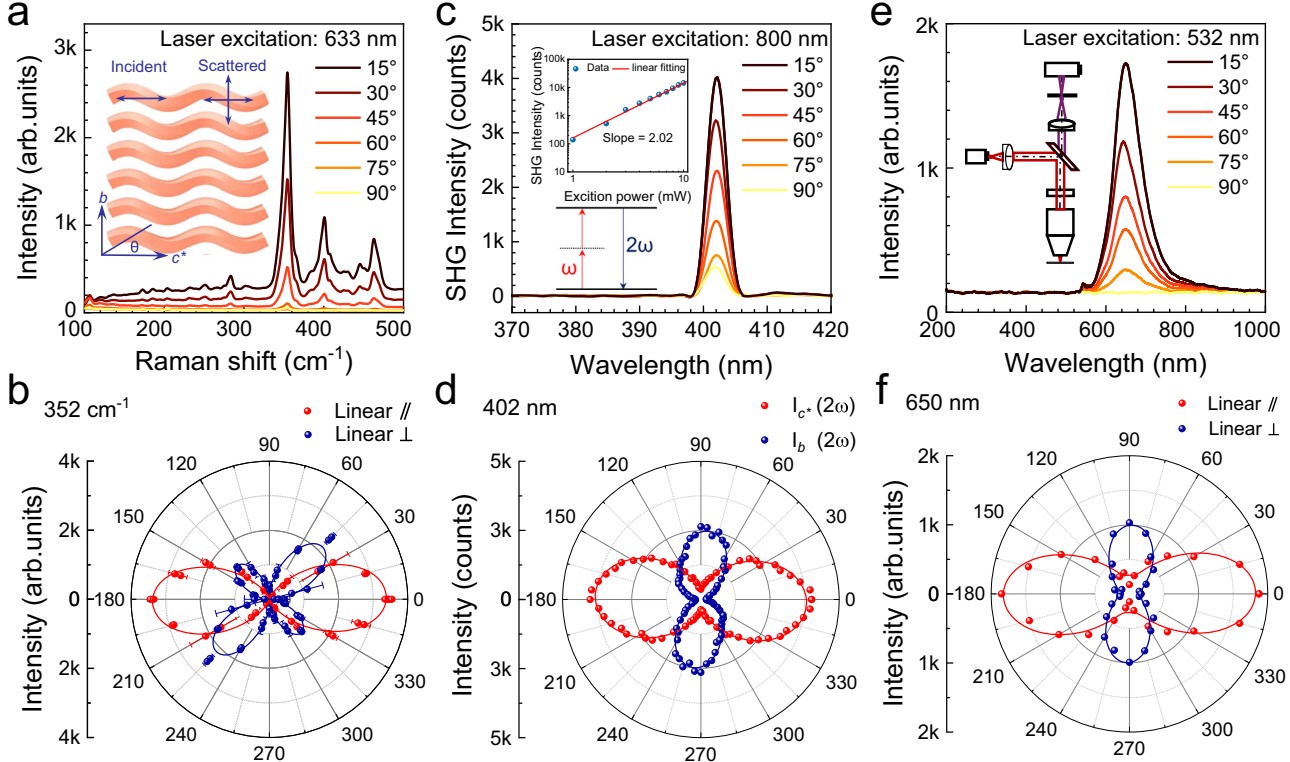

**Fig. 4 | Optical anisotropy of wavy-tube phosphorus single crystal. a** Polarized Raman scattering spectra of wavy-tube phosphorus (wtP) single crystal with typical vibrational peaks at 352 cm⁻¹, 401 cm⁻¹ and 463 cm⁻¹. The polarized laser excites the $b$-$c^*$ plane of wtP in both co- and cross-polarized configurations, with $\theta$ representing the angle between the incident light and the $c^*$-axis. **b** Polar diagrams measured in the co- (red) and cross-polarized (blue) configurations, demonstrating the angular dependence of the 352 cm⁻¹ peak intensity. **c** Polarized second-harmonic genera-tion (SHG) spectra of the wtP crystal. The upper inset illustrates the fitted relationship between the SHG intensity and the excitation light power, while the lower inset depicts the SHG process schematically. **d** Polar diagrams of SHG intensity versus polarization angle $\theta$ for the co- (red) and cross-polarized (blue) configurations. **e** Polarization angle-dependent photoluminescence (PL) spectra of wtP crystals. The inset graph is the testing optical path. **f** Polar diagrams of the angular dependence of 650 nm PL peak intensity spectra tested in co- (red) and cross-polarized (blue) configurations.

## Methods

### Synthesis of 1D phosphorus crystal

The amorphous red phosphorus (aRP, Shanghai Aladdin Biochem-ical Technology Co., Ltd., 99.999%) and tin powder (Sn, Alfa Aesar, 99.995%) were used as the phosphorus source and mineralizer agent, respectively, without additional treatment. The reaction was conducted by a modified chemical vapor phase transport (CVT) method in a dual-temperature zone tube furnace (OTF-1200X-II). Typically, aRP and Sn with a mass ratio of 10:1 were sealed in an ampoule with a length of 100 mm, inner diameter of 18 mm, and thickness of 2 mm at a vacuum of 10⁻⁴ Pa. The temperature of the source zone was raised to above 550 °C to ensure complete sub-limation of aRP. The temperature was then gradually lowered to 460 °C at a rate of 0.06 °C/min for 120 h, followed by a further decrease to 320 °C at a rate of 0.2 °C/min and slow cooling to the ambient temperature for 6 h. Throughout the reaction, a 20 °C temperature difference was maintained between the growth zone and the source zone. The wavy-tube phosphorus (wtP) sample was obtained in the growth zone, and the detailed synthesis is illustrated in Supplementary Information Fig. 2 obtained.

### Optical micrograph and scanning electron microscopy (SEM)

The optical images were obtained on a metallographic microscope (Olympus, BX53) with 5-100x objective lenses and a digital camera. The SEM images and corresponding energy dispersive X-ray spectroscopy (EDS) data were acquired on the ZEISS SUPRA™ 55 (15 kV) equipped with an Oxford X-Max 20 electric-cooled X-ray spectrometer.

### Single crystal X-ray diffraction (SCXRD)

Single crystals with a regular shape and flat shiny surface were selected for structure analysis by XRD using the Rigaku XtaLAB Synergy R dif-fractometer equipped with a hybrid pixel array detector. The multi-layer mirror monochromatic Mo K$\alpha$ ($\lambda$ = 0.71073 Å) radiation was the X-ray source. Data collection, cell refinement, and data reduction were performed by the Rigaku CryAlis PRO program at room temperature. The Olex2 software package was employed to analyze the data[49–51]. The crystal data and structural refinement of the crystal were as follows: monoclinic, space group $P2_1$, $a = 13.0518(3)$ Å, $b = 34.4922(4)$ Å, $c = 18.8538(4)$ Å, $\beta = 109.737(3)°$, $V = 7989.1(3)$ Å³, $Z = 360$, $T = 293$ K, $R_1 = 0.0858$, and $wR_2 = 0.2801$. The X-ray crystallographic coordinates for the structure reported in this study have been deposited at the Cambridge Crystallographic Data Center (CCDC), under deposited number 2322552. These data can be obtained free of charge via www. ccdc.cam.ac.uk/data_request/cif, or by https://doi.org/10.5517/ccdc. csd.cc2hyt1b, or by emailing da-ta_request@ccdc.cam.ac.uk, or by contacting The Cambridge Crystallographic Data Center, 12 Union Road, Cambridge CB2 1EZ, UK; fax: +44 1223 336033.

### Powder X-ray diffraction (PXRD)

The wavy-tube phosphorus crystal flakes were transferred to a diffraction-free substrate to determine the preferred orientation. Some samples were ground in an onyx mortar and sieved to form powder to acquire the non-preferred-orientation PXRD pattern. The PXRD data were collected on an X-ray diffraction instrument (Rigaku Smart Lab 3 kW (40 kV, 30 mA)) using Cu K$\alpha$ radiation ($\lambda$ = 1.5406 Å).

## Ultraviolet photoelectron spectroscopy (UPS)

The UPS measurements were performed on the Nexsa instrument (ThermoFisher) with a low-power Al Kα X-ray source. The sample work function was calculated by subtracting the secondary electron cut-off energy from the optical energy of the He I source (21.22 eV).

## Transmission electron microscopy (TEM)

**Planar TEM samples.** The as-prepared crystal was transferred onto a copper network microgrid with a 200 mesh carbon support film using a dry transfer platform. The grid was subsequently transferred to a double-tilt TEM holder for characterization.

**Cross-sectional TEM sample prepared by focused ion beam (FIB).** The cross-sectional TEM samples were prepared by FIB (Thermo Scientific Scios). A 100 nm carbon layer was deposited on the sample surface to prevent irradiation damage, and tungsten strips 1 μm thick were deposited onto selected locations in specific directions perpendicular and parallel to the long edges of the crystal, where the W strips prevent the crystal from being damaged by the $Ga^+$ beam and provide mechanical support. The milled sample was transferred to a half-copper grid using an Easylift nano-manipulator. Using a thinning stage, the samples were milled by a 30 kV $Ga^+$ beam at currents of 300, 100, 50, and 30 pA sequentially. The possible damaged areas were removed by a 5 kV $Ga^+$ beam at a current of 48 pA for 15 s. Finally, the samples were polished with a 2 kV $Ga^+$ beam at a current of 43 pA.

**TEM.** Cross-sectional TEM images of wtP crystals were characterized employing the Thermo Fisher Talos F200X system at an accelerating voltage of 200 kV. Planar TEM images were characterized with the JEOL-2100Plus system at 200 kV. Scanning transmission electron microscopy (STEM) were carried out on the JEM-ARM 300 F cold-field-emission double spherical aberration-corrected transmission electron microscopy (Cs-corrected TEM) at 300 kV. Aberration-corrected STEM was conducted at a convergence angle of ≈24.5 mrad, and the detection angles were 11–22 mrad and 54–220 mrad.

## Angle-resolved polarized Raman spectroscopy (ARPRS)

A high-resolution confocal Raman instrument (LabRAM HR800 HORIBA JOBIN YVON) with a 633 nm laser excitation was employed to acquire the Raman scattering spectra with a raster line resolution of 1800/mm at room temperature. The ARPR spectra were obtained in the co- and cross-polarized configurations using an acquisition time of 10 s per angle and a laser power of about 10 mW.

## Angle-resolved polarized photoluminescence (PL)

The PL spectra were obtained on the MicOS-iHR320 analyzer (HORIBA) with a 532 nm laser and cryostat. The angle-resolved polarized PL spectra were acquired in the co- and cross-polarized configurations using the 532 nm laser, respectively. The acquisition time was 5 s per angle, and the laser power was about 500 μW.

## Angle-resolved polarized second harmonic generation (SHG)

The SHG spectra were obtained on a reflective configuration microscope system with an 800 nm excitation femtosecond pulse laser (Chamelon Ultra II, repetition frequency 80 MHz). The angle-resolved polarized SHG measurements were conducted in both co-polarized and cross-polarized configurations using an excitation power of 400 μW and a step size of 5°.

## Angle-resolved polarized transmission spectroscopy and Atomic force microscopy

The transmission spectra were acquired on a fully automated micro-area photoelectric test system (MAPS-Mark1-ULRaman-SHG-C). The sample was identical to that used for ellipsometer testing, with polarization testing achieved by rotating the sample. Following optical data acquisition, atomic force microscopy testing of the samples was performed with Cypher S (Asylum Research, USA) devices.

## Polarization-resolved optical microscopy

The polarization-resolved optical images were captured by a metallographic microscope (Olympus BX53) with a polarization system with 20× optical objectives. The images were taken with incident polarized light in the transmission mode perpendicular to the analysis while rotating the sample from 0° to 180° at 15° intervals.

## Spectroscopic ellipsometry measurement

Spectroscopic ellipsometry data were collected using an ME-Mapping-40F ellipsometer (Wuhan Eoptics Technology Co., Wuhan, China) equipped with dual-rotating compensator, with data processing performed via Eometrics software. Experiments were carried out within the wavelength range of 280 nm to 1000 nm, with the incident angle set at 64.95°. Two different orientations of the sample (optical axis parallel and perpendicular) were selected, from which the material's optical constants were calculated via inversion using data within this wavelength band. During the experiment, wtP crystals were dry-transferred onto quartz substrates. Combining an anti-back-reflection optical path with a converging micro-spot eliminated all spurious information from the substrate's back reflection.

## First-principle calculations

The lattice dynamics, electron wave functions, and charge densities of the 1D phosphorus allotrope were computed based on the density-functional theory (DFT) with the generalized gradient approximation (GGA) in the Perdew-Burke-Ernzerhof (PBE) formalism[30]. The calculations were performed using the Vienna Ab initio simulation package (VASP) with a plane-wave basis and a kinetic energy cut-off of 500 eV, as determined by convergence tests. The Brillouin zone (BZ) integration was performed using a Γ-centered $3 \times 3 \times 1$ Monkhorst-Pack k-point grid. The crystal structure was optimized by relaxing the lattice parameters and atomic positions with an energy tolerance of $1 \times 10^{-5}$ and a maximum atomic force tolerance of 0.001 eV/Å. The electron properties were determined by the Heyd-Scuseria-Ernzerhof (HSE06) hybrid exchange-correlation functional[52,53].

In the phonon dispersion calculations, the finite displacement method was employed in conjunction with the Gaussian Approximation Potential (GAP) approach[31]. This method introduced atomic displacements in the optimized crystal structure to compute the interatomic force constants, which were used to construct the dynamical matrix and derive the phonon dispersion curves. The GAP potential was known for its ability to achieve ab initio accuracy with improved computational efficiency, making it suitable for large-scale simulations. The phonon dispersion results were analyzed using the Phonopy software package.

The Raman spectral intensities were determined by the 'Raman-sc' code in combination with Phonopy and VASP. The vibrational properties of orange phosphorus were further determined by considering interlayer interactions within the tube model, which played a pronounced role in its vibrational characteristics. Additionally, Van der Waals corrections were applied to the structure of orange phosphorus to account for long-range interactions. However, these corrections were not included in the phonon dispersion calculations, as the adopted finite displacement method did not support the vdW correction in VASP. The accuracy of the computational results was verified by comparing them with phonon dispersion calculations by the finite-difference method.

## Data availability

Source data are provided with this paper.

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

## Acknowledgements

This work was supported by the National Key R&D Program of China (2023YFA0915600 (Jiahong Wang)), the National Natural Science Foundation of China (52272268 (S.F.J.), 62405352 (S.Z.)), the Guang-dong Basic and Applied Basic Research Foundation (2025B1515020088 (Jiahong Wang), 2024A1515030176 (Jiahong Wang), 2024A1515012644 (S.Z.)), Guangdong Provincial Key Laboratory of Multimodality Non-Invasive Brain-Computer Interfaces (2024B1212010010 (Jiahong Wang)), Shenzhen Science and Technology Program (RCJC20200714114435061 (X.-F.Y.), JCYJ20220818100806014 (Jiahong Wang)), Postdoctoral Fellowship Program of CPSF (GZC20232869 (S.Z.)), City University of Hong Kong Donation Research Grants (DON-RMG 9229021 and 9229021 (P.K.C.)) and the research council of Finland Flagship Program (320167, PREIN (Z.P.S.)). We also appreciate the support and assistance of Wuhan Eoptics Technology Co. and R&D Center of Waynelabs Instruments & Solutions, Hubei Zhongwei Optoelectronic Technology Co., Ltd. in the field of optical characterization testing. This work is part of the Finnish Center of Excellence in Quantum Materials (QMAT).

## Author contributions

S.Z. synthesized crystals, analyzed crystal structures and collected optical data. Z.L.L. collected and interpreted single-crystal structures. T.T.J., M.Q.F., and L.Y. assisted with the crystal growth experiments. M.Q.F., C.W. and Jiahui Wang performed photoluminescence and transmission spectral polarization experiments. H.W. and Y.L. assisted with Raman spectral polarization experiments and analyses. M.Q.F. and S.Z. conducted the spectroscopic ellipsometry measurement. L.P.D. calculated and extracted the lattice structure information and the optical constants. Y.Y. performed second harmonic generation experiments. X.D.H., S.F.M., and B.S.X. assisted with the TEM characterization analysis. X.L.C. and C.Y. assisted with the data analysis. X.F.C. and P.K.C. revised the manuscript. S.F.J. guided the crystal structure characterization analysis. F.D. provided and guided the first-principles calculations. X.-F.Y., Z.P.S. and Jiahong Wang contributed to the conception of the research. S.Z. and Jiahong Wang conceived and coordinated the research. S.Z. analyzed the data and wrote the manuscript with revision from Jiahong Wang; All authors discussed the results and reviewed the manuscript.

## Competing interests

The authors declare no competing interests.

## Additional information

[1]Shenzhen Institutes of Advanced Technology, Chinese Academy of Sciences, Shenzhen, PR China. [2]School of Integrated Circuits, Dongguan University of Technology, Dongguan, PR China. [3]Beijing National Laboratory for Condensed Matter Physics, Institute of Physics, Chinese Academy of Sciences, Beijing, PR China. [4]University of Chinese Academy of Sciences, Beijing, PR China. [5]Southern University of Science and Technology, Shenzhen, PR China. [6]Materials Institute of Atomic and Molecular Science, Shaanxi University of Science and Technology, Xi'an, PR China. [7]Department of Optoelectronic Science & Technology, School of Electronic Information and Artificial Intelligence, Shaanxi University of Science & Technology, Xi'an, China. [8]Research Division of Advanced Materials, Suzhou Laboratory, Suzhou, China. [9]College of Electronic Information and Optical Engineering, Taiyuan University of Technology, Taiyuan, PR China. [10]School of Science and Technology, Nottingham Trent University, Nottingham, UK. [11]Department of Physics, Department of Materials Science and Engineering, and Department of Biomedical Engineering, City University of Hong Kong, Tat Chee Avenue, Kowloon, Hong Kong, China. [12]Key Laboratory of Biomedical Imaging Science and System, Chinese Academy of Sciences, State Key Laboratory of Biomedical Imaging Science and System, Shenzhen, PR China. [13]Department of Electronics and Nanoengineering, Aalto University, Espoo, Finland. [14]Guangdong Provincial Key Laboratory of Multimodality Non-Invasive Brain-Computer Interfaces, Shenzhen, PR China. [15]These authors contributed equally: Shuai Zhang, Zhaolong Liu. ✉e-mail: scu_ding@163.com; f.ding@siat.ac.cn; xf.yu@siat.ac.cn; zhipei.sun@aalto.fi; jh.wang1@siat.ac.cn

