## [Peer Review File · Nature Communications]

Strong optical anisotropy in one-dimensional phosphorus wavy tubes

Corresponding Author: Professor Zhipei Sun

Version 0:

Reviewer comments:

Reviewer #3

(Remarks to the Author)

The authors synthesized a one-dimensional phosphorus single crystal composed of wavy polygonal tube phosphorus and extracted its birefringent complex refractive index using spectroscopic ellipsometry. The birefringence is as high as 0.95 in the visible range and is maintained higher than 0.5 across the visible and near-infrared range (up to 800 nm), whereas that in the previous result (published in Nature Photonics) was as high as 0.63 in a narrower wavelength range (~650 nm).

My initial concerns regarding this manuscript were:

(1) whether a birefringence of 0.63 is high enough compared to previous papers, considering that several studies have reported birefringence values higher than 1 in the visible range. In the revised manuscript, the birefringence is as high as 0.95 and is maintained higher than 0.5 across the visible and near-infrared range (up to 800 nm).

(2) the authors extracted the refractive index using angle-dependent reflection measurements, but there were significant issues—for example, they did not consider optical interference effects in the films. In the revised manuscript, they utilized spectroscopic ellipsometry to extract the refractive index, and thus, the authors rigorously considered the optical interference effects.

The authors addressed all of my initial concerns, and therefore, I recommend considering this manuscript for publication in Nature Communications, after addressing a minor question.

I recall that the birefringence in the authors' previous work submitted to Nature Photonics was as high as 0.63 and was maintained only around ~650 nm. However, in the revised manuscript, the birefringence is significantly increased across a much wider wavelength range. Could the authors explain the origin of such a large difference? Does this difference arise from the initial assumptions in the angle-dependent reflection measurements, such as neglecting optical interference effects in the films?

Response to Referees

We thank the reviewers for their helpful suggestions and constructive comments. We have addressed all the concerns raised by the reviewers. The paper (NCOMMS-25-97677-T) has been accordingly revised.

For clarity, revisions in the manuscript are marked in **red**, and our responses are provided in **blue**. In the following, we address the reviewers' comments one by one.

Reviewer #3:

General comment

The authors synthesized a one-dimensional phosphorus single crystal composed of wavy polygonal tube phosphorus and extracted its birefringent complex refractive index using spectroscopic ellipsometry. The birefringence is as high as 0.95 in the visible range and is maintained higher than 0.5 across the visible and near-infrared range (up to 800 nm), whereas that in the previous result (published in Nature Photonics) was as high as 0.63 in a narrower wavelength range (~650 nm).

My initial concerns regarding this manuscript were:

(1) whether a birefringence of 0.63 is high enough compared to previous papers, considering that several studies have reported birefringence values higher than 1 in the visible range. In the revised manuscript, the birefringence is as high as 0.95 and is maintained higher than 0.5 across the visible and near-infrared range (up to 800 nm).

(2) the authors extracted the refractive index using angle-dependent reflection measurements, but there were significant issues—for example, they did not consider optical interference effects in the films. In the revised manuscript, they utilized spectroscopic ellipsometry to extract the refractive index, and thus, the authors rigorously considered the optical interference effects.

The authors addressed all of my initial concerns, and therefore, I recommend considering this manuscript for publication in Nature Communications, after addressing a minor question.

Our response

We thank the reviewer for the positive assessment and recommendation. We are pleased that we have addressed all the initial concerns. We have carefully addressed the remaining question in the revised manuscript and the following response letter, and we have updated the relevant text accordingly.

Comment 1

I recall that the birefringence in the authors' previous work submitted to Nature Photonics was as high as 0.63 and was maintained only around ~650 nm. However, in the revised manuscript, the birefringence is significantly increased across a much wider wavelength range. Could the authors explain the origin of such a large difference? Does this difference arise from the initial assumptions in the angle-dependent reflection measurements, such as neglecting optical interference effects in the films?

Our response

We thank the reviewer for this important question. The difference mainly comes from how thin-film interference was treated in the earlier analysis versus the revised manuscript. In short, we now determine the refractive index using spectroscopic ellipsometry with an interference-aware anisotropic thin-film model, which provides accurate and physically consistent values that are directly comparable to, and consistent with, established literature. Details are as follows.

In our earlier (preliminary) work, we extracted optical constants from angle-dependent reflectance and applied Kramers–Kronig analysis, assuming the measured reflectance $R_{\text{meas}}(\lambda)$ was close to the intrinsic reflectance of the material. However, our film thickness (~200 nm) lies in the coherent regime where Fabry–Pérot interference strongly modulates the measured reflectance. Because these interference fringes were not explicitly modelled, part of the thickness-induced oscillations was incorrectly attributed to intrinsic spectral features. This leads to systematic errors in the extracted refractive indices (especially the phase information) and therefore underestimates n and the birefringence Δn , producing artificially narrow, peak-like birefringence features around limited wavelengths.

In the revised manuscript, we remeasured the same type of samples using spectroscopic ellipsometry, which directly measures Ψ and Δ and explicitly accounts for thin-film interference. We fit the data with a model that simultaneously optimizes film thickness and the full anisotropic optical-constant tensor. This separates intrinsic material dispersion from geometric interference effects and enables reliable birefringence extraction over a broad wavelength range.

Note that the ellipsometry technique is the established standard for characterizing anisotropic thin films, and its validity is supported by recent high-impact studies [e.g., Nat. Photonics, 12(7), 392-396 (2018); Nat. Photonics, 18(9), 922-927 (2024); arXiv preprint arXiv:2412.12697 (2024)]. Therefore, all optical parameters in the revised manuscript are obtained using this rigorous method, ensuring a fair and precise comparison with existing literatures

Our modification to the manuscript:

(Line 219, page 22: in the revised Supplementary Information)

“Conventionally, the optical properties of materials are extracted from reflectance

spectra ($R(\omega)=r_0^2(\omega)$) combined with the Kramers-Kronig (*K-K*) relations. In such method, the phase shift $\theta(\omega_0)$ is derived to reconstruct the complex refractive index using the integral:

$$\theta(\omega_0)=-\frac{\omega_0}{\pi}P\int_0^\infty\frac{\ln R(\omega_0)}{\omega^2-\omega_0^2}d\omega$$

where ω is the angular frequency, and P denotes the Cauchy principal value. And the material's optical constants yielding:

$$n(\omega)=\frac{1-r_0^2(\omega)}{1+r_0^2(\omega)-2r_0(\omega)\cos\theta(\omega)}$$

$$\kappa(\omega)=\frac{2r_0(\omega)\sin\theta(\omega)}{1+r_0^2(\omega)-2r_0(\omega)\cos\theta(\omega)}$$

However, this approach faces significant challenges when applied to thin films due to the Fabry-Pérot interference effect, governed by the condition:

$$2nd\cos\phi=m\lambda$$

where n is the refractive index, d is the film thickness, ϕ is the angle of refraction, λ is the wavelength, and m is an integer representing the interference order. In transparent or weakly absorbing regions, this interference can cause reflectance to fluctuate drastically between 10% and 90%, depending on the refractive index contrast. Consequently, applying K-K analysis introduces significant artifacts, with errors in the extracted absorption coefficient (α), and deviations in the refractive index (n).

In contrast, spectroscopic ellipsometry avoids these limitations by measuring the complex reflectance ratio ρ , defined as:

$$\rho=\frac{r_p}{r_s}=\tan(\Psi)e^{i\Delta}$$

Here, r_p and r_s are the complex reflection coefficients for parallel and perpendicular polarizations, respectively. By simultaneously measuring two independent parameters—the amplitude ratio (Ψ) and the phase difference (Δ)—this technique allows for the construction of an optical model where the film thickness d is treated as an independent fitting variable. This capability effectively decouples geometric interference effects from the intrinsic dielectric functions, yielding accurate optical constants. As a result, we use spectroscopic ellipsometry in this work to determine optical parameters.”